# The Role of Surgical Axillary Staging Prior to Immediate Breast Reconstruction in the Era of De-Escalation of Axillary Management in Early Breast Cancer

**DOI:** 10.3390/jpm12081283

**Published:** 2022-08-04

**Authors:** Miriam Svensson, Looket Dihge

**Affiliations:** 1Department of Surgery, Kristianstad Central Hospital, SE-291 33 Kristianstad, Sweden; 2Department of Clinical Sciences, Division of Surgery, Lund University, SE-221 85 Lund, Sweden; 3Department of Plastic and Reconstructive Surgery, Skåne University Hospital, SE-214 28 Malmö, Sweden

**Keywords:** breast cancer, axillary lymph node status, sentinel lymph node biopsy, de-escalating, radiotherapy, mastectomy, immediate breast reconstruction

## Abstract

Postmastectomy radiotherapy (PMRT) following immediate breast reconstruction (IBR) is associated with postoperative complications. Although the incidence of node-positive breast cancer is declining, a separate sentinel lymph node biopsy (SLNB) is still performed before mastectomy when IBR is planned, in order to evaluate nodal status and the need for PMRT. This study assessed the impact of staged SLNB on the breast reconstructive planning, and presents common clinicopathological characteristics of breast cancer with macrometastatic nodal spread where staged SLNB would be beneficial to indicate PMRT. Medical records of breast cancer patients scheduled for mastectomy and IBR at Skåne University Hospital, Sweden, from November 2014 to February 2020, were reviewed. Of 92 patients, node-positive disease was present in 15 (16%). Fifty-three patients underwent staged SLNB before mastectomy and IBR, and 10 (19%) presented with nodal metastasis. All patients with macrometastatic sentinel nodes were presented with palpable, multifocal, ER+ breast carcinoma of no special type with tumor size > 17.0 mm. Overall, four women received PMRT after verified metastasis by staged SLNB, and IBR was cancelled for three patients. These findings question the benefit of routine staged SLNB before mastectomy and IBR in breast cancer populations within established mammography screening programs with low risk of nodal metastasis.

## 1. Introduction

Immediate breast reconstruction (IBR) improves the quality of life of women with breast cancer undergoing mastectomy [1], and should be offered to the vast majority of these patients, according to European guidelines [2]. For thosereceiving IBR, postmastectomy radiation therapy (PMRT) is widely known to cause an increased risk of postoperative complications [3,4,5], and IBR is usually deferred when PMRT is recommended. PMRT reduces the risk of recurrence and breast cancer mortality in patients with node-positive (N+) disease [6], and is recommended when lymphatic macrometastasis (>2.0 mm) is present [2,7]. The difficulty of predicting the patient’s lymph node status preoperatively, and, thus, any need for radiation therapy, complicates individualized risk assessment for women desiring IBR.

Today, sentinel lymph node biopsy (SLNB) is the gold standard for the evaluation of axillary lymph node status in patients with clinically node-negative (cN0) disease. Previous studies suggest SLNB to be performed as a separate surgical axillary staging procedure, prior to the final decision on the suitability of IBR [8]. However, a staged SLNB entails an extra operation for the patient, increases the risk of infection, extends the time to cancer surgery in the breast, and increases hospital costs [9]. To evaluate the appropriate timing of SLNB (i.e., staged or at the same time of mastectomy), the prevalence of patients benefitting from an altered treatment planning due to lymphatic metastasis displayed in staged SLNB must be considered.

Since the introduction of a public mammography screening service, the prevalence of breast cancer patients presenting with heavy burden lymphatic metastasis at the time of diagnosis has declined [10,11], and the importance of PMRT for breast cancer patients with minor metastatic burden has been questioned [12]. For patients with only sentinel lymph node (SLN) micrometastasis (≤2 mm), PMRT is today not routinely recommended [2,7]. Consequently, only a minority of newly diagnosed breast cancer patients opting for IBR are likely to be recommended PMRT based on nodal metastasis displayed in SLNB, and the benefit from routine staged SLNB to predict PMRT before IBR is, thus, questionable. Nevertheless, there is a paucity of contemporary data on axillary lymph node status in breast cancer patients within an established mammography screening program undergoing staged SLNB prior to mastectomy and planned IBR.

The primary aim of this study was to assess axillary lymph node involvement in breast cancer patients scheduled for mastectomy and IBR, and to determine nodal characteristics of harvested SLNs to evaluate the necessity of routinely staged SLNB to indicate PMRT and influence breast reconstruction options. A secondary aim was to present common features of early breast cancer presenting with macrometastatic nodal disease at the time of diagnosis, where SLNB would be beneficial to predict the need for PMRT.

## 2. Materials and Methods

### 2.1. Study Population

Patients with cN0 primary invasive breast cancer or suspected microinvasive ductal carcinoma in situ (miDCIS), scheduled for surgical axillary nodal staging, mastectomy, and IBR at Skåne University Hospital, Sweden, from November 2014 to February 2020 were identified. Both patients scheduled for primary mastectomy and patients scheduled for completion mastectomy after breast conserving therapy (BCT) due to inadequate surgical margins were included. The choice of mastectomy primarily was based on preoperatively assessed tumor size, including the extent of DCIS, tumor location in the breast, any instance of heredity breast cancer, and/or patient’s preference. Exclusion criteria were: men, neoadjuvant chemotherapy, previous axillary surgery due to a history of invasive breast cancer or breast carcinoma in situ, omission of surgical axillary staging, or missing data on nodal status. Data on histopathological characteristics of the primary tumor and harvested lymph nodes, patient characteristics, and surgical method of the breast and axilla were obtained from medical records. Bilateral breast cancer was considered as two separate cases. PMRT was routinely recommended for patients with axillary macrometastasis, along with completion axillary lymph node dissection (cALND), according to Swedish National Guidelines for Breast Cancer [13]. Patients with SLN micrometastasis or ≤2 SLN macrometastases were offered participation in the SENOMIC and SENOMAC trials, respectively [14,15], which allocated patients to either undergo cALND or no further axillary surgery. The study was approved by the Swedish Ethical Review Authority (2020-01967).

### 2.2. Evaluation of Histological and Immunohistochemical Characteristics

Histopathological and immunohistochemical parameters were obtained by breast pathologists by preoperative evaluation of core needle biopsy (CNB) and postoperative evaluation of the excised section. Histological type, nuclear grade of DCIS, and histological grade of invasive tumors were evaluated according to the Swedish Society of Pathology criteria [16]. No distinction was made between multicentric and multifocal tumors in this study; multifocality was defined as more than one tumor focus within the same breast, separated by benign tissue. Expression levels of estrogen receptors (ER) and progesterone receptors (PR) were assessed by immunohistochemistry (IHC) and considered positive if ≥1 percent positive nuclei staining, according to ESMO definitions [2]. Human epidermal growth factor receptor 2 (HER2) status was assessed by IHC and completion in situ hybridization (ISH). Values of Ki-67 were interpreted according to the local cut-off of the laboratory of Skåne University Hospital, and values >20 percent were defined as high. Classification of molecular surrogate subtype was made according to the definitions of the 2019 St Gallen consensus [17,18,19].

### 2.3. Evaluation of Axillary Lymph Node Status

All included patients underwent preoperative clinical examination of the axilla and routine axillary ultrasound (AUS) as part of the diagnostic work-up. For women with abnormal clinical status or AUS imaging, complementary fine-needle aspiration biopsy (FNAB) was performed. Patients with normal clinical assessment of the axilla and normal AUS or FNAB were considered cN0. Final axillary lymph node staging was assessed by SLNB or SLNB and cALND. According to the American Joint Committee on Cancer (AJCC) classification criteria, a node-positive status was assigned if axillary lymph nodes harbored micrometastases (>0.2 mm or >200 cells and ≤2.0 mm) or macrometastases (>2.0 mm) [20]. Lymph nodes with isolated tumor cells (ITC) (≤0.2 mm or ≤200 cells) were considered node-negative (N0).

### 2.4. Statistical Analysis

Descriptive statistics were applied to display patient and tumor characteristics and mode of tumor detection (mammography screening versus symptomatic) in the overall study cohort. The study cohort was dichotomized into those undergoing primary mastectomy and those undergoing completion mastectomy after BCT due to inadequate surgical margins. The distribution of clinicopathological characteristics and nodal status across the two groups were compared using Pearson χ^2^ test, chi-squared test for trend, Fisher’s exact test for categorical variables, and Mann–Whitney for continuous variables. All statistical tests were 2-tailed, and *p* < 0.05 was considered statistically significant. Statistical analyses were computed in SPSS^®^, statistics for Windows, version 26.0 (IBM^®^ Corp., Armonk, NY, USA).

## 3. Results

### 3.1. Study Population

Between November 2014 and February 2020, 109 women were diagnosed with cN0 primary breast cancer or suspected miDCIS, and scheduled for mastectomy and IBR at Skåne University Hospital, Sweden (Figure 1). Of these, 13 patients were excluded due to neoadjuvant chemotherapy. Two patients were excluded due to omission of SLNB, and two were excluded due to failure to identify any lymph node at routine surgical axillary staging. In total, 92 patients constituted the overall study cohort.

### 3.2. Patient and Tumor Characteristics

In the overall study cohort, 59 patients underwent primary mastectomy and 33 patients underwent completion mastectomy after BCT due to inadequate surgical margins. Patient and tumor characteristics for all included patients and the overall study cohort dichotomized into those who underwent mastectomy and those who underwent completion mastectomy after BCT are presented in Table 1. The preoperatively assessed median tumor extent was significantly larger for those who underwent primary mastectomy (50.0 mm) compared with those who underwent completion mastectomy after BCT (23.0 mm) (*p* = 0.001). There was, however, no significant difference in largest invasive tumor size on the final histopathological examination (*p* = 0.626). For age, mode of detection, histopathological tumor characteristics, axillary lymph node status, and type of performed IBR, no statistically significant differences are identified across the two groups (primary mastectomy versus completion mastectomy after BCT due to inadequate surgical margins).

### 3.3. Axillary Lymph Node Status and Nodal Characteristics

Axillary lymph node metastasis was displayed in 15 of 92 patients (16 percent) (Table 2). In one of the patients, micrometastatic deposit was found in the concomitantly excised non-SLN. SLN macrometastasis was displayed in 3 of 15 patients with N+ disease. For all three, a solitary lymph node excised at SLNB harbored metastatic deposit classified as macrometastasis, while other nodes harbored micrometastatic disease. When examining lymph nodes harvested at ALND, only one additional metastatic lymph node was presented in one patient. Accounting for final axillary lymph node status after cALND, the largest metastatic deposit for this patient was converted from micrometastatic to macrometastatic. In total, lymphatic macrometastasis was found in four patients after surgical axillary staging by SLNB and cALND.

For the 53 patients who underwent staged SLNB prior to mastectomy, lymphatic metastasis was found in 10 (19 percent). cALND was performed in one of the eight patients displaying only SLN micrometastasis, and displayed one further metastatic lymph node with macrometastatic deposit. All other patients with SLN micrometastasis were enrolled in the SENOMIC trial [14], with no further axillary surgery.

SLN macrometastasis was detected in 2 of 53 patients who underwent SLNB as a separate surgical axillary staging procedure prior to scheduled mastectomy and IBR. Both patients refrained from cALND and enrollment in SENOMAC [15].

### 3.4. Impact of Staged SLNB on PMRT and Immediate Breast Reconstruction

Four women received PMRT after verified SLN metastasis by staged SLNB, and for all four, node-positive status was the only indication for irradiation (Figure 2). Two of the women were diagnosed with SLN macrometastasis. For one patient, macrometastatic deposit was found in a lymph node harbored during cALND, and for another patient, only micrometastatic nodal deposits were displayed.

IBR with tissue expander was performed in one patient with lymphatic metastasis receiving PMRT on the patient’s request, albeit with information on risk of complications related to irradiation in combination with breast reconstruction. For the other three patients presenting with SLN metastasis, and who were candidates for PMRT, IBR was deferred.

### 3.5. Features of Breast Cancer with Lymph Node Macrometastasis in SLNB

All patients with macrometastatic deposit in SLNB had palpable, multifocal, ER+ primary breast cancer tumors of no special type (NST). The postoperative invasive tumor size was >17.0 mm for SLN macrometastatic breast cancer. Although all patients with SLN macrometastasis were within the age range of the public mammography screening program, two out of three patients had symptomatically discovered tumors. Lymphovascular invasion (LVI) of the primary tumor was found in all patients except for one.

## 4. Discussion

In the current study, the incidence of nodal metastasis for patients with primary breast cancer scheduled for mastectomy and IBR was 16 percent. Macrometastatic deposit in SLNB was found in 3 of 92 patients, and IBR was deferred in 3 of 53 patients based on SLN metastasis by staged SLNB. Thus, separate nodal staging by staged SLNB prior to mastectomy and IBR have only marginal impact on the breast reconstructive planning. The vast majority of the patients diagnosed with breast cancer or suspected miDCIS scheduled for mastectomy and IBR do not benefit from an altered treatment planning based on the outcome of staged SLNB. To the best of our knowledge, this is the only updated study assessing axillary lymph node status and nodal characteristics in a contemporary population of breast cancer patients scheduled for mastectomy and IBR within an established mammography screening service.

Along with tumor size, nodal status is recognized as one of the most important predictors of PMRT [21]. In the current study, preoperative examination of axillary lymph node status before scheduled mastectomy and IBR involves clinical examination and AUS. While AUS is implemented as part of the routine diagnostic work-up and enables non-invasive preoperative evaluation of axillary nodal status, it is an unreliable staging modality for patients presenting with low axillary nodal metastatic burden [12]. The high false negative rate of AUS in identifying N+ breast cancer signifies the technical limitations of preoperative imaging in accurately predicting nodal status and need for PMRT.

For patients receiving IBR, PMRT is associated with a complication rate of 18–73 percent [3,22], and a risk of reconstruction failure of 25 percent [3]. Implant-based reconstructions are particularly associated with a higher risk of complications, including capsular contracture and loss of implant, with the highest complication rate detected for patients receiving tissue expanders [4]. For those receiving autologous breast reconstruction, PMRT is associated with increased rates of autologous fat necrosis and fibrosis [5]. Regardless of reconstructive method, an increased risk for postoperative infections is displayed.

SLNB enables a complete pathological evaluation of harvested lymph nodes. Compared with axillary dissection, postoperative arm morbidity is reduced [23] with preserved oncosurgical safety [24]. Thus, SLNB remains the gold standard for axillary staging in patients with cN0 disease. However, there is no consensus regarding timing of the SLNB procedure in breast cancer patients scheduled for mastectomy and IBR.

Intraoperative cytopathological analysis of SLNs enables a perioperative evaluation of axillary nodal status, and can be used to indicate need for PMRT [25]. However, concern has been raised over the risk of false negative results [26]. In addition to being an expensive and time-consuming diagnostic tool, the estimated sensitivity for detecting axillary metastasis by intraoperative cytopathological analysis is inferior to the standard pathological embedding of lymph nodes, and ranges from 57 to 73 percent [27,28]. Although intraoperative pathological examination of SLNs is more reliable for detecting macrometastases [27], many breast centers still prefer separate SLNB when mastectomy and IBR is planned [29].

In a previously performed study, endorsing staged SLNB prior to mastectomy and IBR, the option of breast reconstruction was changed for 62 percent of all women with a positive lymph node in staged SLNB [8]. The overall prevalence of N+ disease among patients scheduled for mastectomy and IBR is estimated to be 27 percent. In another similar study, exploring the utility of staged SLNB for patients with breast cancer desiring IBR, the prevalence of N+ tumors is estimated to be 26 percent [30]. For 79 percent of these patients, IBR is deferred due to a positive outcome in staged SLNB.

In this contemporary study cohort, only 3 of 92 patients are diagnosed with SLN macrometastasis, and only one patient displays four or more metastatically involved lymph nodes. According to European guidelines, neither cALND nor locoregional radiotherapy is recommended for breast cancer patients displaying only SLN micrometastasis when no other indication is present (i.e., T3-T4 tumors and/or involved resection margins) [2,13]. For patients with SLN macrometastasis, ESMO guidelines endorse PMRT in patients displaying four or more axillary metastatic lymph nodes, whereas it should be considered in patients with one to three metastatically involved lymph nodes [7]. Similarly, a majority of the panelists of the 2019 St Gallen Consensus Conference [19] favored PMRT in patients with one to three metastatic lymph nodes only with coexisting adverse features (i.e., TNBC or HER2+ breast cancer, tumor size >5 cm, and patients with residual disease after neoadjuvant chemotherapy). Consequently, few patients in the present study with planned mastectomy and IBR meet the criteria for PMRT based on the nodal status outcome of SLNB alone, and IBR is deferred in only three patients displaying a positive lymph node in staged SLNB.

The low number of nodal metastasis, and particularly the low rate of SLN macrometastasis found in our study, may be explained by the well-established Swedish National Breast Cancer Screening program, which is available for all Swedish women within the age range of 40–74 years [10,11]. Mammography screening enables the detection of breast cancer at an early stage before palpation is possible. As tumor size is recognized to be closely associated with axillary lymphatic metastasis [31,32], breast cancer screening by mammography influences the prevalence of patients presenting node-positive disease at the time of diagnosis [33]. In the present study, most patients have screening-detected tumors, and display primary breast cancer with a small median invasive tumor size of 13.0 mm in the overall study cohort. Moreover, all patients with SLN macrometastasis have palpable, multifocal breast cancer with invasive tumor size > 17,0 mm, and two of three patients have symptomatically discovered tumors.

The low prevalence of nodal spread may also be explained by a high rate of patients displaying DCIS in final histopathological evaluation. It must be noted that the absence of microinvasive foci cannot be determined solely by preoperative CNB. A meta-analysis shows that about one in four diagnoses of DCIS at CNB represent under-staged invasive breast cancer [34]. The prognostic implications of microinvasion in DCIS has been investigated by Magnoni et al. [35]. In their study, only 2 percent of patients with miDCIS display macrometastasis in SLNB. This result is in accordance with the findings from the present study, in which all patients with SLN macrometastasis display invasive breast cancer of NST in final histopathological examination. The long-term outcomes from their study show a low rate of regional recurrence in patients with miDCIS and positive SLNs. They, therefore, suggest SLNB to be of little benefit for patients with suspected miDCIS, due to low risk of nodal metastasis and overall good prognosis.

This study has several limitations. Besides its retrospective nature, the study population comprised a single-center cohort. On the other hand, Skåne University Hospital constitutes one of the largest centers for breast cancer in Sweden, counting the number of diagnosed patients. Furthermore, the study cohort is defined by the consecutive inclusion of patients during a period of more than five years, and all variables are obtained from medical records and not registries, which minimizes the potential impact of incomplete data. Moreover, by excluding patients undergoing neoadjuvant therapy, the confounding influence of this parameter on the evaluation of nodal status and clinicopathological characteristics of the primary tumor could be minimized.

## 5. Conclusions

This study shows that staged SLNB prior to mastectomy and IBR in early breast cancer may not add significant clinical value for the prediction of node-positive axillary status and probability of PMRT. Although PMRT following IBR is associated with a high complication rate, the prevalence of patients who would benefit from staged SLNB prior to mastectomy and IBR is low. In particular, the number of patients displaying macrometastatic deposit in SLNB is low. Our findings, thus, question the benefit of routine staged SLNB prior to scheduled mastectomy and IBR in patient populations within an established public mammography screening service with low risk of lymphatic metastasis. The results support the evidence that less axillary surgery can provide the same level of information for the pre-operative planning of breast reconstruction in early breast cancer.

## Figures and Tables

**Figure 1 jpm-12-01283-f001:**
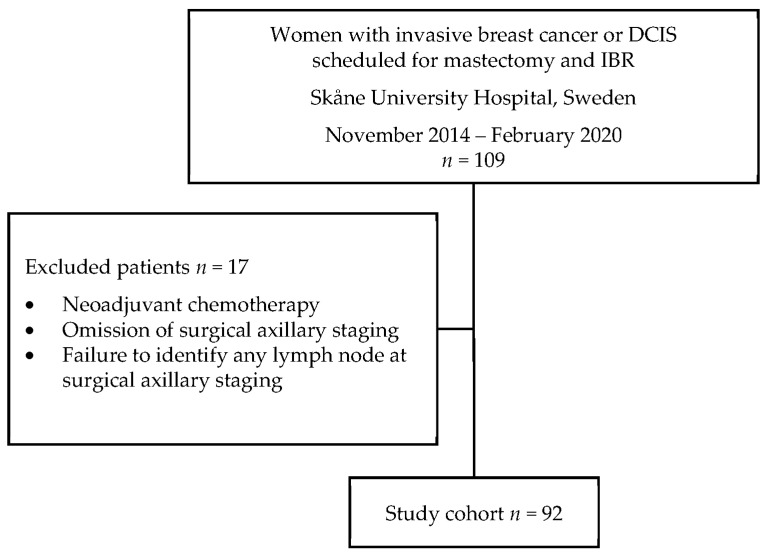
Flow chart of original patient population, excluded patients, and selected study cohort. Abbreviations: *DCIS,* ductal carcinoma in situ; *IBR,* immediate breast reconstruction.

**Figure 2 jpm-12-01283-f002:**
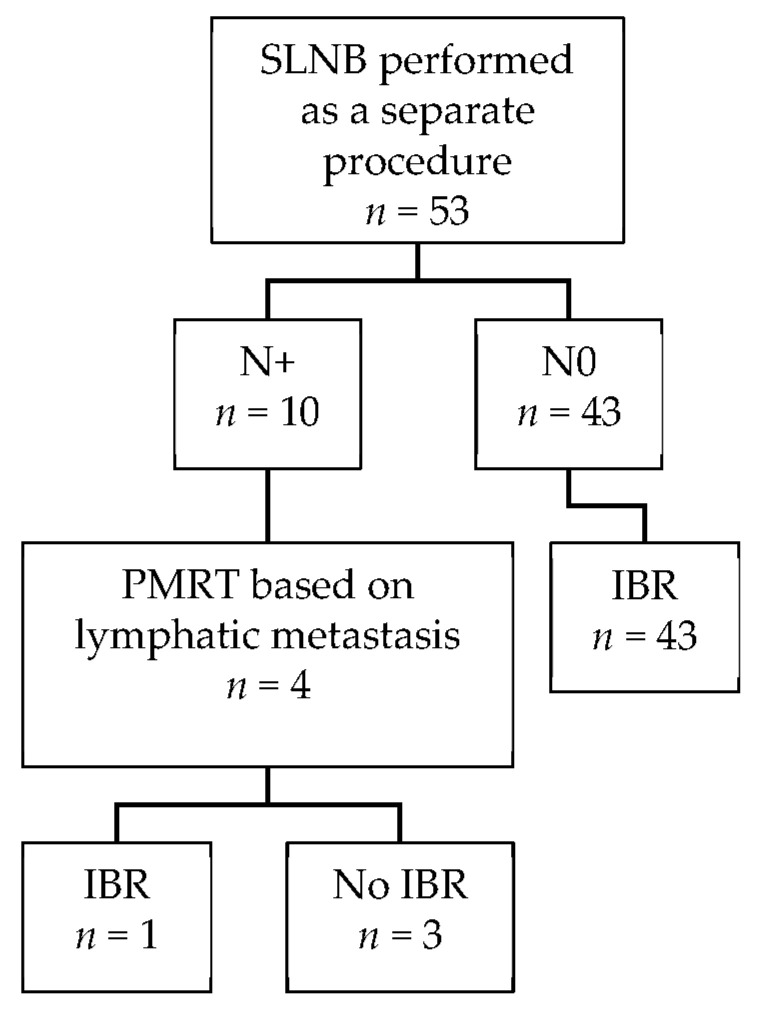
Flow chart of patients who underwent sentinel lymph node biopsy (SLNB) as a separate axillary staging procedure, patients who receive postmastectomy radiation therapy (PMRT) based on lymphatic metastasis in staged SLNB, and number of patients who receive immediate breast reconstruction (IBR) or no IBR. Abbreviations: *SLNB*, sentinel lymph node biopsy; N+, node-positive; N0, node-negative; PMRT, postmastectomy radiation therapy; IBR, immediate breast reconstruction.

**Table 1 jpm-12-01283-t001:** Patient and tumor characteristics in the overall study cohort.

Variables	All *n* = 92	Primary Mastectomy *n* = 59	Completion Mastectomy after BCT *n* = 33	*p* Value
Age *, years	49 (23–74)	48 (23–74)	51 (29–71)	0.077 ^a^
Mode of detection ^ꝉ^				0.181 ^b^
Symptomatic	32 (41)	21 (45)	11 (34)	
Screening	47 (60)	26 (55)	21 (66)	
Preoperative tumor characteristics				
Estimated tumor extent *, mm	40.0 (4.0–100.0)	50.0 (4.0–100.0)	23.0 (12.0–70.0)	0.001 ^a^
Missing	11	7	4	
Postoperative tumor characteristics				
DCIS	37 (40)	23 (39)	14 (42)	0.747 ^b^
Nuclear grade				0.871 ^d^
I	3 (8)	2 (9)	1 (7)	
II	13 (35)	9 (39)	4 (29)	
III	21 (57)	12 (52)	9 (64)	
Invasive breast cancer	55 (60)	36 (61)	19 (58)	0.747 ^b^
Histological type				0.862 ^d^
NST	44 (80)	28 (78)	16 (84)	
Lobular	8 (15)	6 (17)	2 (11)	
Other	3 (5)	2 (6)	1 (5)	
Nottingham histological grade				0.284 ^c^
I	14 (26)	8 (22)	6 (35)	
II	30 (57)	21 (58)	9 (53)	
III	9 (17)	7 (19)	2 (12)	
Missing	2	0	2	
ER status				0.594 ^d^
Negative	4 (7)	2 (6)	2 (11)	
Positive	50 (93)	34 (94)	16 (89)	
Missing	1	0	1	
PR status				0.701 ^d^
Negative	9 (17)	7 (19)	2 (11)	
Positive	45 (83)	29 (81)	16 (89)	
Missing	1	0	1	
HER2 status				1.000 ^d^
Non-amplified	49 (93)	33 (92)	16 (94)	
Amplified	4 (8)	3 (8)	1 (6)	
Missing	2	0	2	
Ki-67 *, %	24 (4–59)	24 (4–59)	21 (6–50)	0.252 ^a^
Missing	2	1	1	
Surrogate molecular subtype				0.213 ^d^
Luminal A-like	18 (35)	10 (29)	8 (47)	
Luminal B-like/HER2-negative	27 (53)	20 (59)	7 (41)	
Luminal B-like/HER2-positive	3 (6)	3 (9)	0	
HER2-positive/non-luminal	1 (2)	0	1 (6)	
TNBC	2 (4)	1 (3)	1 (6)	
Missing	4	2	2	
Largest tumor size *, mm	13.0 (0.2–55.0)	14.0 (1.3–55.0)	11.0 (0.2–31.0)	0.626 ^a^
Multifocality				0.868 ^b^
No	24 (44)	16 (44)	8 (42)	
Yes	31 (56)	20 (56)	11 (58)	
Lymphovascular invasion				1.000 ^d^
Absent	38 (72)	25 (71)	13 (72)	
Present	15 (28)	10 (29)	5 (28)	
Missing	2	1	1	
Axillary lymph node status				0.560 ^d^
N0	77 (84)	48 (81)	29 (88)	
N+	15 (16)	11 (19)	4 (12)	
Type of immediate breast reconstruction				
IBR deferred	3 (3)	2 (5)	1 (2)	0.314 ^d^ (type of implant not included)
DIEP-flap	11 (12)	7 (17)	4 (8)	
Implant	78 (85)	33 (79)	45 (90)	
Expander	55 (71)	23 (70)	32 (71)	
Permanent	23 (30)	10 (30)	13 (29)	

Values in parentheses are valid percentages of each column if not otherwise explained. The percentage values are rounded and total percentage may, therefore, not be 100. *p* values are calculated for patients undergoing primary mastectomy versus patients undergoing completion mastectomy after breast conserving therapy (BCT) due to inadequate surgical margins. * Median (range). ^ꝉ^ Only patients within the age range of 40–74 years, for whom participation in the Swedish National Breast Cancer Screening program is offered (*n* = 79). ^a^ Mann–Whitney; ^b^ Pearson χ^2^ test; ^c^ chi-squared test for trend; ^d^ Fisher’s exact test. Abbreviations: BCT, breast conserving therapy; DCIS, ductal carcinoma in situ; NST, invasive carcinoma of no special type; ER, estrogen receptor; PR, progesterone receptor; HER2, human epidermal growth factor receptor 2; TNBC, triple-negative breast cancer; N0, lymph node-negative; N+, lymph node-positive; IBR, immediate breast reconstruction; DIEP-flap, deep inferior epigastric perforator-flap.

**Table 2 jpm-12-01283-t002:** Axillary lymph node status and nodal characteristics.

Variables	All *n* = 92	SLNB Performed as a Separate Procedure *n* = 53
SLNB		
No. lymph nodes excised, median (range)	2 (1–7)	2 (1–7)
Lymph node status by SLNB, *n* (%)		
N0	77 (84)	43 (81)
N+	15 (16)	10 (19)
Micrometastasis	12 (80)	8 (80)
Macrometastasis	3 (20)	2 (20)
No. metastatic lymph nodes in N+ patients, median (range)	1 (1–4)	1 (1–4)
No. Micrometastases	1 (0–3)	1 (0–3)
No. Macrometastases	0 (0–1)	0 (0–1)
Largest metastatic deposit, mm, median (range)	1.40 (0.25–10.00)	0.85 (0.25–10.00)
Missing	1	0
Completion ALND		
N (%)	4 (4)	1 (2)
No. lymph nodes excised, median (range)	16 (11–25)	11
No. metastatic lymph nodes, median (range)	0 (0–1)	1
SLNB + completion ALND		
N0	77 (84)	43 (81)
N+	15 (16)	10 (19)
Micrometastasis	11 (73)	7 (70)
Macrometastasis	4 (27)	3 (30)

Values in parentheses are valid percentages of each column if not otherwise explained. The percentage values are rounded and total percentage may, therefore, not be 100. Abbreviations: *SLNB*, sentinel lymph node biopsy; N0, node-negative; N+, node-positive; ALND, axillary lymph node dissection.

## Data Availability

The data that support the findings of this study may be made available from the corresponding author upon reasonable request.

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
