# Peer review of "The Role of Surgical Axillary Staging Prior to Immediate Breast Reconstruction in the Era of De-Escalation of Axillary Management in Early Breast Cancer"

_jpm, 2022, doi:10.3390/jpm12081283_

Round 1
Reviewer 1 Report
Congratulations for your good job in this manuscript.
However, some minor issue may confuse the readers
1. Line 140~141, the total number of “patients” presented DCIS (47) and invasive breast cancers (55) is not equal to the enrolled “patients” in this study (92)
2. As you mentioned in the discussion, relatively small sample size of this study is a critical flaw. Especially comparing to the reference cited (#8. McGuire K et al. Timing of sentinel lymph node biopsy and reconstruction for 385 patients undergoing mastectomy. Ann Plast Surg. 2007;59(4):359-63.)
3. It is an interesting issue about the effectiveness and timing of axillary surgery for breast cancer patients. Several new studies and literatures can be found. However, some of the references cited in this article seemed to be out of date (published for more than 10 years). As the authors found and mentioned in the discussion section, the declined prevalence of positive node may be explained by the well-established Swedish National Breast Cancer Screening program which is available for all Swedish women within the age range of 40-74 years. Comparing the present study with those studies in recent years with improved breast cancer survey program will be more convincing.
Reviewer 2 Report
Dear authors, I would like you revise the following points:
page 2, line 61, is access right? or it must be “assess”
page 2, line 74, non-radical breast conserving therapy, in my opinion this term should be modified; I suppose that patients did not receive radiotherapy, instead you may use breast conserving surgery, I suppose that the problem was inadequate surgical margins.
I recommend you describe that in easy way explaining the cause of mastectomy (inadequate surgical margins after breast conserving surgery)
Pages 4 to 8, if you use text the tables are not necessaries; I recommend use Tables to expose results
Why do you split the series in primary mastectomy or completion mastectomy after non-radical BCT; is it useful? There are not differences.
You end point is that only 4% of the all patients had macro metastasis- it implies a radiotherapy recommendation- so separate SLNB is unnecessary. In my opinion, this conclusion can be refuted; do use SLNB in the surgical procedure of mastectomy to indicate an axillary lymphadenectomy in case of macro metastasis detected by intraoperative pathological study or not and you decide it after final pathological report?
If you use SLNB with intraoperative pathological study, you can detect macro metastasis and decide about IBR. Obviously, separate SLNB has a low profit but in my opinion, it has not sense because you do it in unique surgical procedure including mastectomy and IBR.
Round 2
Reviewer 2 Report
Dear authors, I appreciate your modifications, the article can be published in this form. I would just like to mention that in our breast unit we start with the axillary procedure (sentinel node identification) and we send the axillary tissue for an intraoperative anatomopathological study, which usually takes about 40 minutes; the case of false positive is very rare, while we perform the mastectomy and when we receive the report we decide on lymphadenectomy and breast reconstruction.
This manuscript is a resubmission of an earlier submission. The following is a list of the peer review reports and author responses from that submission.